# Factors Affecting Taiwanese Consumers' Intention to Purchase Abnormally Shaped Produce

**Yen-Lun Su [1], Pongsakorn Ngamsom [2],* and Jen-Hung Wang [3]**

[1] Department of Hotel and Restaurant Management, National Pingtung University of Science and Technology, Pingtung 91201, Taiwan; allansu@mail.npust.edu.tw

[2] Department of Tropical Agriculture and International Cooperation, National Pingtung University of Science and Technology, Pingtung 91201, Taiwan

[3] Rdata System Co., Ltd., Taichung 406047, Taiwan; a82277186@gmail.com

* Correspondence: p10422017@mail.npust.edu.tw; Tel.: +886-90969-6368

**Abstract:** This study examines the effects of produce shape abnormality, organic labeling, and discounts on consumers' intention to purchase produce. Two studies were conducted. In Study 1, a 3 (normal, moderate, and extreme shape abnormality) × 2 (with and without organic labeling) between-subjects design was used. In Study 2, a 2 (moderate and extreme shape abnormality) × 5 (discount: 30%, 40%, 50%, 60%, and 70%) between-subjects design was used. A total of 480 valid responses to questionnaires were collected. Study 1 revealed that the interaction between produce shape abnormality and organic labeling significantly affects purchase intention. Study 2 revealed that the interaction between produce shape abnormality and discount significantly affected purchase intentions. For a 30% discount, the results indicate no difference in intention to purchase moderately or extremely abnormally shaped produce. For 40%, 50%, 60%, and 70% discounts, intention to purchase moderately abnormally shaped produce was stronger than that for extremely abnormally shaped produce. This study discusses the implications of the findings, limitations, and recommendations for future research.

**Keywords:** produce shape abnormality; organic labeling; discount; purchase intention

## 1. Introduction

Food waste is a crucial topic because of its negative effects on the environment. Studies have indicated that nearly one-third of food produced worldwide is wasted every year at various stages of the food supply chain and by consumers [1–3]. Food waste threatens global food security, contributes to greenhouse gas emissions and soil depletion, and causes price inflation [3,4]. Fruits and vegetables are frequently wasted and comprise approximately 50% of household waste [5]. Produce waste primarily occurs when food still suitable for human consumption is discarded [6]. Another primary cause of such waste is aesthetic objections to fruits and vegetables that appear abnormal or have an odd shape [7]. Porter et al. [8] estimated that aversion to abnormal produce accounts for the loss of one-third of farm production in Europe [8]. To reduce food waste, retailers and local markets in Taiwan have offered abnormally shaped produce since 2015, but the public's acceptance of such products remains undetermined.

Most studies on imperfect food have been conducted in America and Europe [3,7,9–13]. Studies in Taiwan have examined several topics related to suboptimal food, including appearance, business models, risk perception, and purchase intention [14–18]. However, no study has specifically analyzed the effects of intrinsic and extrinsic cues on consumers' intention to purchase abnormally shaped food in Taiwan. Studies have reported that consumers are more willing to purchase discounted abnormally shaped produce [19,20]. Studies have recommended investigating the effect of price on customers' willingness to purchase abnormally shaped produce because price should determine consumers' attitudes toward abnormal food [7,21]. Because food waste at the retail and consumption stages is

increasing in Taiwan, understanding the factors that affect consumers' intention to purchase abnormally shaped produce is imperative.

This study investigates whether food shape abnormality and organic labeling influence consumers' purchase intention and the extent to which discounts affect purchase intention. The results can provide reference for managers and policymakers developing interventions to encourage the consumption of abnormally shaped produce and reduce waste.

## 2. Literature Reviews

### 2.1. Produce Shape Abnormality and Purchase Intention

Studies have demonstrated that the appearance of food, which includes its shape, color, and presentation, determines consumers' initial sensory impressions and results in inferences of quality; these impressions then influence consumers' preferences and purchase intention [22–24]. Food retailers generally reject imperfect produce because of the assumption that consumers are unwilling to purchase food that visually deviates from the norm; this aversion results from the perception that food abnormalities are associated with lower product quality [13,25]. A Danish study conducted by Loebnitz et al. [7] indicated that extremely abnormal food shape affects consumers' purchase intention but moderately abnormal food shape does not.

Loebnitz and Grunert [7,22] investigated the effects of produce shape abnormality on the purchase intention of consumers in China [25]. Their results indicated that food shape influenced purchase intention and that consumers were more likely to purchase normally shaped produce than moderately or extremely abnormally shaped produce. Su et al. [18] surveyed 400 Taiwanese consumers, and the results indicated that they had the strongest intention to purchase normally shaped food and that purchase intentions decreased as food shape abnormality increased. Consumers should exhibit a stronger intention to purchase normally shaped produce than moderately or extremely abnormally shaped produce. On this basis, this study proposes the following hypothesis:

**Hypothesis 1 (H1).** *Produce shape abnormality affects purchase intention; consumers exhibit lower intention to purchase moderately and extremely abnormally shaped produce than to purchase normally shaped produce.*

### 2.2. Effects of Organic Labeling on Purchase Intention

Consumers evaluate products by analyzing product cues, which in turn influence purchase intentions. Olson's cue utilization theory states that consumers make inferences about product quality through intrinsic and extrinsic cues, which then determine purchase decisions [26]. Intrinsic cues are the physical attributes of a product, such as its ingredients and shape. Extrinsic cues refer to attributes that are not part of the physical product, such as brand name and price [7]. This study examines the effects of an intrinsic cue, namely shape abnormality, and an extrinsic cue, organic labeling, on consumers' purchase intention.

Organic foods are produced without genetic modification or synthetic chemicals, such as pesticides and fertilizers [27]. Organic certification is a credence quality; therefore, trust in organic integrity is crucial to ensuring consumers buy organic products [28,29]. Because most consumers lack the technical expertise, knowledge, and resources to identify organic foods, specific labels are used [29,30]. Thus, organic labeling constitutes an extrinsic cue to consumers [7,31].

Numerous studies have examined consumers' intention to purchase organic products. Michaelidou and Hassan [32] discovered that the primary motivations to purchase organic products include social image, price, quality, and health and safety. Studies have indicated that the major reasons customers purchase organic foods are concerns about health, food safety, taste, the environment, animal welfare, and a desire to support the local economy [33–36]. A literature review of organic food consumption conducted by Hemmerling et al. [37] identified health, taste, safety, and the environment as the principal motivations for purchasing organic foods. Concerns about food safety, health, and the environment are all

related to consumer involvement with organic foods [32,34–37]. An empirical study in Taiwan revealed that health-conscious respondents had a stronger intention to engage in health-promoting behavior and consume organic food [29]. In addition, the study revealed an association between concerns about food safety and the purchasing of organic foods. Respondents who prefer chemical-free, natural, and safe foods are more likely to purchase organic food [38–40]. Considering the increase in health-conscious Taiwanese consumers concerned about food safety, this study proposes the following hypothesis:

**Hypothesis 2 (H2).** *Consumers' intention to purchase organic products is stronger than that for nonorganic products.*

### 2.3. Interaction between Produce Shape Abnormality and Organic Labeling

Cue utilization theory states that cues can have predictive value (PV) and confidence value (CV). PV is the degree to which consumers link a certain cue to quality, which indicates the reliability of the cue. CV indicates the confidence of consumers in accurate judgment and cue utilization. PV and CV play a critical role in evaluation. Intrinsic cues are more reliable than extrinsic cues. However, when intrinsic cues are unavailable, PV and CV are low, and consumers tend to rely on extrinsic cues [41].

Cue diagnosis theory posits that the effect of a cue depends on cue diagnosis; negative cues yield more diagnostic power because of consumers' negative bias [42]. Produce shape abnormality constitutes a special cue because of its rarity in stores. Organic labeling is not a special cue because consumers often encounter it on food, clothing, and personal care products. Produce shape abnormality and organic labeling both provide diagnostic cues to consumers but with varying levels of specificity [7]. Purohit and Srivastava [43] classified cues as high and low scope. High-scope cues are established over time and do not change suddenly. Low-cope cues are temporary and can change. Produce shape abnormality is a high-scope cue because the quality cue of physical appearance does not change quickly; organic labeling is a low-scope cue.

Studies have noted that intrinsic cues (e.g., food shape abnormality) are stronger than extrinsic cues (e.g., organic labeling) and that negative assumptions drawn from high-scope cues weaken the diagnostic power of low-scope cues [7,43]. On this basis, this study proposes the following hypothesis:

**Hypothesis 3 (H3).** *The interaction of produce shape abnormality and organic labeling affects consumers' purchase intention; when produce is normally (abnormally) shaped and labeled as organic (not labeled as organic), purchase intention increases (decreases).*

### 2.4. Effects of Discount on Purchase Intention

Promotions, a crucial marketing tool, have garnered the attention of both practitioners and researchers. Discounts are the most prevalent type of promotion in the service industry. Discounts can encourage purchase and product trials [44]. Studies have revealed a significant, positive relationship between discount and purchase intention [45,46]. Wan et al. [47] indicated that discounts affect consumers' willingness to repurchase a product and that brand image moderates the relationship between discount and consumers' willingness to repurchase a product.

Petruzzelli [48] revealed a positive correlation between willingness to purchase low-grade produce and discount; 56% of respondents were willing to purchase free, low-grade produce. Nusair et al. [49] indicated that price affects consumers' purchase intention. Their results also indicate that willingness to purchase a discounted service decreases after 80% discounts. Consumers exhibited a stronger intention to purchase discounted services offered by quick-service restaurants and budget hotels. The study identified the most appropriate discount for quick-service restaurants and outlet mall services to be no more than 60%. On this basis, this study hypothesizes the following:

**Hypothesis 4 (H4).** *The interaction of produce shape abnormality and discount affects consumers' purchase intention; when produce is normally (abnormally) shaped and sold at a high (low) discount, purchase intention increases (decreases).*

## 3. Study 1

### 3.1. Methods

#### 3.1.1. Participants

For Study 1, a 3 (produce shape abnormality: normal, moderate, and extreme) × 2 (with and without organic labeling) between- and within-subjects design was employed. The study randomly recruited consumers from supermarkets and traditional markets in the Changhua and Pingtung counties of Taiwan. The data were collected over a period of two weeks during September 2021. The researchers proceeded with the data collection during normal supermarket and traditional market operating hours, consisting of approximately two hours of data collection per day during the morning, afternoon, and evening in order to try to ensure the randomness of the sample population. The participants were randomly assigned to one of six conditions and asked to describe their intention to purchase produce by scanning a quick response (QR) code with their mobile phones. After invalid data were removed, a total of 180 valid questionnaire responses remained for analysis. The participants were 65% women and ranged in age from 41 to 60 years. They were relatively well-educated, with 53.9% possessing an associate or bachelor's degree.

#### 3.1.2. Stimuli and Variables

This study used images of an apple, lemon, carrot, and eggplant, each at three levels of shape abnormality, a total of 12 images (Figure 1; developed by Leobnitz et al., 2015) [7]. The food in half of the images had organic labeling. This study subjected all images to a manipulation check before purchase intention was measured to determine whether the normal, moderate, and extreme levels of shape abnormality were sufficiently different and to ensure the organic labels would be recognized. The study measured purchase intention using a 7-point Likert-type scale developed by Loebnitz et al. [7]. Purchase intention was assessed using the question "How likely would you be to purchase these food items?", with 1 indicating "very unlikely" and 7 indicating "very likely". The questionnaire was developed in English, translated into Chinese and back-translated to ensure quality. An example of the questionnaire is included in Appendix A.

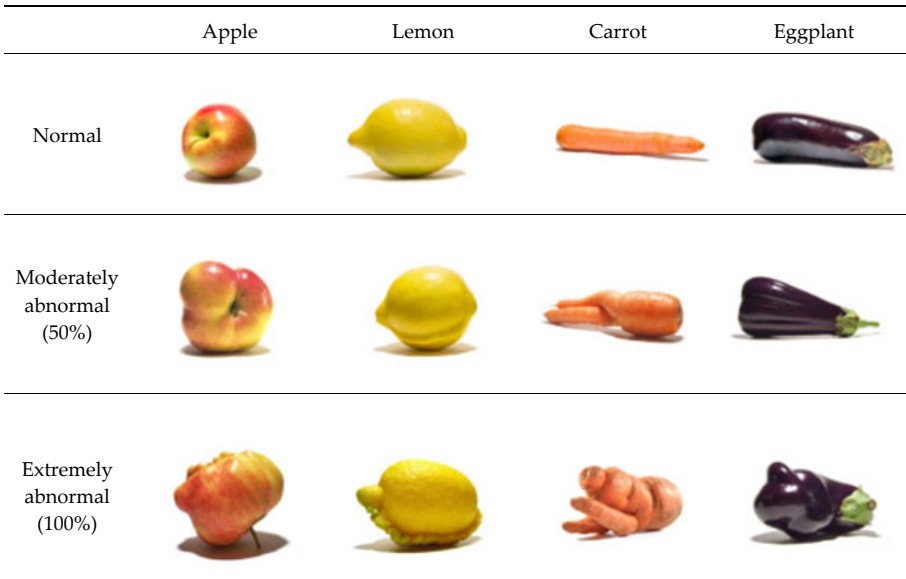

**Figure 1.** Experimental stimuli.

*3.2. Results*

3.2.1. Manipulation Check

This study assessed the manipulation of produce shape abnormality by instructing the participants to indicate how normal (abnormal) the food appeared on a 3-point scale (1 = "normal", 2 = "moderately abnormal", 3 = "extremely abnormal"). To determine the participants' familiarity with organic labeling, the participants were instructed to indicate whether they recognized the labels ("yes"/"no"). This study excluded questionnaires with answers that were inconsistent with the depicted picture or label to ensure successful manipulation.

3.2.2. Hypothesis Testing

The means (Ms) and standard deviations (SDs) of intention to purchase among the six conditions are presented in Table 1. A two-way analysis of variance [50] was performed to test the research hypotheses (Table 2). The results indicate a significant main effect for produce abnormality (F = 19.996, $p < 0.001$). Intention to purchase normal produce (M = 5.22, SD = 1.35) was significantly higher than intention to purchase moderately abnormal (M = 4.50, SD = 1.65, $p < 0.05$) and extremely abnormal (M = 3.40, SD = 1.86) produce ($p < 0.01$). The mean intention to purchase moderately abnormal produce (M = 4.50, SD = 1.65) was higher than that for intention to purchase extremely abnormal produce (M = 3.43, SD = 1.86; $p < 0.001$). These results support Hypothesis 1, that intention to purchase moderately and extremely abnormally shaped produce would be weaker than intention to purchase normally shaped produce.

**Table 1.** Means and standard deviations of purchase intention [a].

| Shape | Organic Labeling | Mean | Std. Deviation | N |
|---|---|---|---|---|
| Normal | Yes | 5.53 | 1.57 | 30 |
| | No | 4.90 | 1.03 | 30 |
| | Total | 5.22 | 1.35 | 60 |
| Moderately abnormal | Yes | 4.70 | 1.60 | 30 |
| | No | 4.30 | 1.70 | 30 |
| | Total | 4.50 | 1.65 | 60 |
| Extremely abnormal | Yes | 4.30 | 1.93 | 30 |
| | No | 2.57 | 1.33 | 30 |
| | Total | 3.43 | 1.86 | 60 |
| Total | Yes | 4.84 | 1.77 | 90 |
| | No | 3.92 | 1.69 | 90 |
| | Total | 4.38 | 1.79 | 180 |

[a] Purchase intention is based a 7-point Likert-type scale (1 = "very unlikely", 7 = "very likely").

**Table 2.** Tests of between-subjects effect-dependent variable: purchase intention for study 1.

| Source | Type III Sum of Squares | df | Mean Square | F | *p*-Value |
|---|---|---|---|---|---|
| Corrected Model | 150.117 [a] | 5 | 30.023 | 12.425 | 0.000 |
| Intercept | 3458.450 | 1 | 3458.450 | 1431.310 | 0.000 |
| Food shape abnormality | 96.633 | 2 | 48.317 | 19.996 | 0.000 |
| Organic labeling | 38.272 | 1 | 38.272 | 15.839 | 0.000 |
| Shape × Organic | 15.211 | 2 | 7.606 | 3.148 | 0.045 |
| Error | 420.433 | 174 | 2.416 | | |
| Total | 4029.000 | 180 | | | |
| Corrected Total | 570.550 | 179 | | | |

[a]: $R^2 = 0.263$ (adjusted $R^2 = 0.242$).

The results indicate significant main effect for organic labeling (F = 15.84, *p* < 0.001; Table 2). Consumers' intention to purchase organic produce was significantly stronger (M = 4.84, SD = 1.77) than their intention to purchase nonorganic produce (M = 3.92, SD = 1.69), supporting Hypothesis 2.

The interaction between produce shape abnormality and organic labeling significantly affected purchase intention (F = 3.148, *p* < 0.05; Table 2), supporting Hypothesis 3. Consumers' intention to purchase normal and moderately abnormally shaped organic produce was not significantly higher than that for nonorganic produce. Consumers' intention to purchase extremely abnormally shaped organic produce was significantly higher (M = 4.30, SD = 1.93) than that for nonorganic produce (M = 2.57, SD = 1.33, *p* < 0.001; Figure 2).

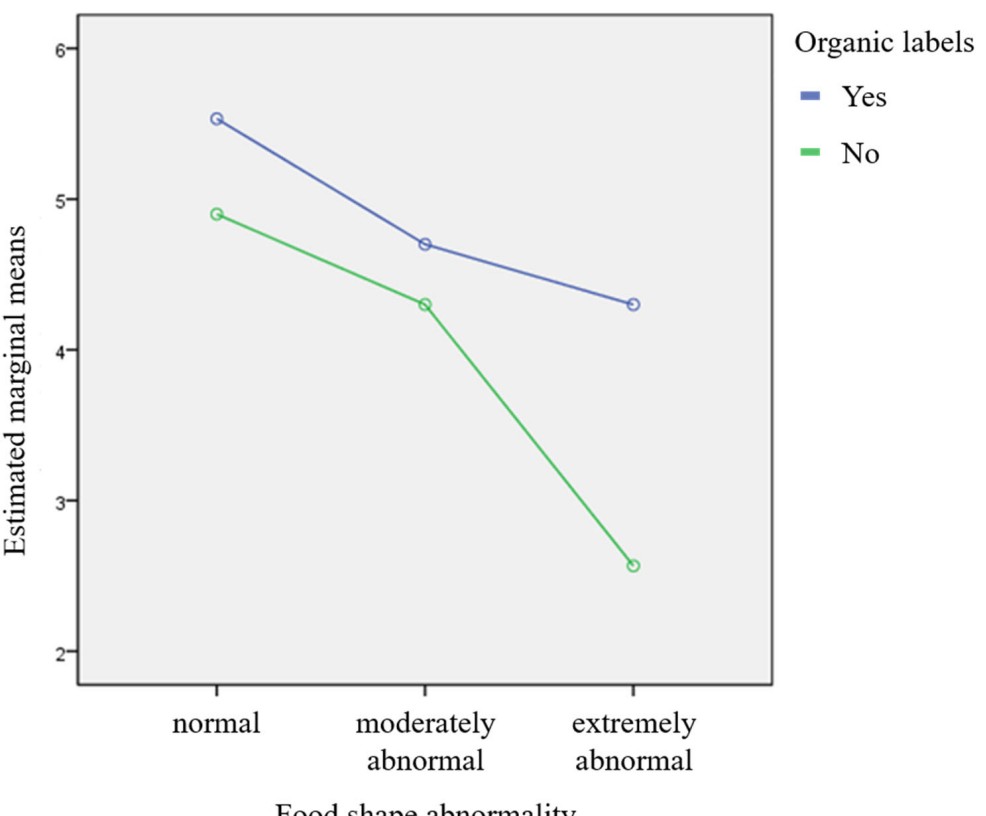

**Figure 2.** Effects of produce shape abnormality and organic labeling on purchase intention.

Intention to purchase normal organic produce (M = 5.53, SD = 1.57) was significantly higher than that for extremely abnormal organic produce (M = 4.30, SD = 1.93, *p* < 0.05). However, the results indicated no significant difference in intention to purchase between normal and moderately abnormal products. In addition, no significant difference in purchase intention was observed between moderately abnormal and extremely abnormal products. Intention to purchase normal nonorganic produce (M = 4.90, SD = 1.03) was significantly higher than that for extremely abnormal nonorganic produce (M = 2.57, SD = 1.33, *p* < 0.05). Intention to purchase moderately abnormal produce (M = 4.30, SD = 1.71) was significantly higher than that for extremely abnormal produce (M = 2.57, SD = 1.33, *p* < 0.05). However, the results indicate no significant difference in purchase intention between normal and moderately abnormal produce.

## 4. Study 2

*4.1. Methods*

4.1.1. Participants

For Study 2, a 2 (produce shape abnormality: moderate or extreme) × 5 (discount: 30%, 40%, 50%, 60%, and 70%) between- and within-subjects design was employed. The study randomly recruited 300 participants from local supermarkets or traditional markets. A questionnaire with one of 10 scenarios was randomly distributed to each participant. The participants were instructed to scan a QR code to complete the questionnaire on their mobile phones. The data were collected over a period of two weeks during October 2021. The researchers proceeded with the data collection during normal supermarket and traditional market operating hours, consisting of approximately two hours of data collection per day during the morning, afternoon, and evening in order to try to ensure the randomness of the sample population.

In each condition, the participants were instructed to express their intention to purchase moderately abnormal ($n = 150$, 30% = 30, 40% = 30, 50% = 30, 60% = 30, 70% = 30) and extremely abnormal ($n = 150$, 30% = 30, 40% = 30, 50% = 30, 60% = 30, 70% = 30) produce. The sample comprised 231 women (77.0%), and most participants (68.7%) were between 41 and 60 years of age. Approximately 59% of the respondents had a high school education or lower.

4.1.2. Stimuli and Variables

Study 2 involves images of an apple, lemon, carrot, and eggplant, with two levels of produce shape abnormality each. The images were modified from those developed by Leobnitz et al. [7]. This study subjected all images to a manipulation check before purchase intention was measured to determine whether the normal, moderate, and extreme levels of shape abnormality were sufficiently different. This study measured intention to purchase produce at five discount levels (30%, 40%, 50%, 60%, and 70%) by using a 7-point Likert-type scale modified from Loebnitz et al. [7]. Purchase intention was measured using the question "How likely would you be to purchase these food items?", with 1 indicating "very unlikely" and 7 indicating "very likely". The questionnaire was developed in English, translated into Chinese, and back-translated to ensure quality. An example of the questionnaire is included in Appendix A.

*4.2. Results*

4.2.1. Manipulation Check

To ensure the successful manipulation of produce shape abnormality, this study instructed the participants to rate how abnormal each food appeared on a 2-point scale (1 = "moderately abnormal", 2 = "extremely abnormal"). To assess the manipulation of discounts, the participants identified discounts on a 5-point scale (1 = 30%, 2 = 40%, 3 = 50%, 4 = 60%, 5 = 70%). The examination results indicate the successful manipulation of both produce shape abnormality and discounts.

4.2.2. Hypothesis Testing

The means (Ms) and standard deviations (SDs) of intention to purchase among the 10 conditions are presented in Table 3.

The interaction between produce shape abnormality and discount significantly affected purchase intention (F = 5.661, $p < 0.001$; Table 4), supporting Hypothesis 4. Intention to purchase moderately abnormal produce at 70% (M = 5.97, SD = 1.22), 60% (M = 5.50, SD = 1.31), and 50% (M = 5.63, SD = 1.03) discounts was significantly higher than intention to purchase moderately abnormal produce at 40% (M = 3.47, SD = 1.61), and 30% (M = 2.53, SD = 0.97) discounts (Figure 3). However, the results indicate no significant difference in purchase intention among produce at 50%, 60%, and 70% discounts. In addition, no significant difference in purchase intentions was observed among produce at 30% and 40% discounts. Intention to purchase extremely abnormal produce at 70% (M = 4.67, SD = 1.35)

and 60% (M = 4.10, SD = 1.47) discounts was significantly higher than intention to purchase such produce at 40% (M = 2.67, SD = 0.96) and 30% (M = 2.67, SD = 1.52) discounts. However, the data indicate no significant difference in purchase intention between produce at a 50% discount and produce at any other discount. In addition, no significant difference in purchase intention was observed between 70% and 60% discounts or between 40% and 30% discounts.

**Table 3.** Means and standard deviations of purchase intention [a] for food shape abnormality and discount level.

| Shape | Discount (% off) | Mean | Std. Deviation | N |
|---|---|---|---|---|
| Moderately abnormal | 30 | 2.53 | 0.97 | 30 |
| | 40 | 3.47 | 1.61 | 30 |
| | 50 | 5.63 | 1.03 | 30 |
| | 60 | 5.50 | 1.31 | 30 |
| | 70 | 5.97 | 1.22 | 30 |
| | Total | 4.62 | 1.84 | 150 |
| Extremely abnormal | 30 | 2.67 | 1.52 | 30 |
| | 40 | 2.67 | 0.96 | 30 |
| | 50 | 3.67 | 1.21 | 30 |
| | 60 | 4.10 | 1.47 | 30 |
| | 70 | 4.67 | 1.35 | 30 |
| | Total | 3.55 | 1.52 | 150 |
| Total | 30 | 2.60 | 1.27 | 60 |
| | 40 | 3.07 | 1.38 | 60 |
| | 50 | 4.65 | 1.49 | 60 |
| | 60 | 4.80 | 1.55 | 60 |
| | 70 | 5.32 | 1.43 | 60 |
| | Total | 4.09 | 1.77 | 300 |

[a] Purchase intention is measured on a seven-point Likert scale (1 = very unlikely, 7 = very likely).

**Table 4.** Tests of between-subjects effect-dependent variable: purchase intention for study 2.

| Source | Type III Sum of Squares | df | Mean Square | F | *p*-Value |
|---|---|---|---|---|---|
| Corrected Model | 458.013 [a] | 9 | 50.890 | 30.892 | 0.000 |
| Intercept | 5010.253 | 1 | 5010.253 | 3041.390 | 0.000 |
| Food shape abnormality | 85.333 | 1 | 85.333 | 51.800 | 0.000 |
| Discount level | 335.380 | 4 | 83.845 | 50.897 | 0.000 |
| Shape × Discount | 37.300 | 4 | 9.325 | 5.661 | 0.000 |
| Error | 477.733 | 290 | 1.647 | | |
| Total | 5946.000 | 300 | | | |
| Corrected Total | 935.747 | 299 | | | |

[a]: $R^2$ = 0.489 (adjusted $R^2$ = 0.474).

Intention to purchase moderately abnormal produce was higher than that for extremely abnormal produce at 70%, 60%, 50%, and 40% discounts. The results indicate no significant difference in purchase intention between moderately and extremely abnormal produce at a 30% discount.

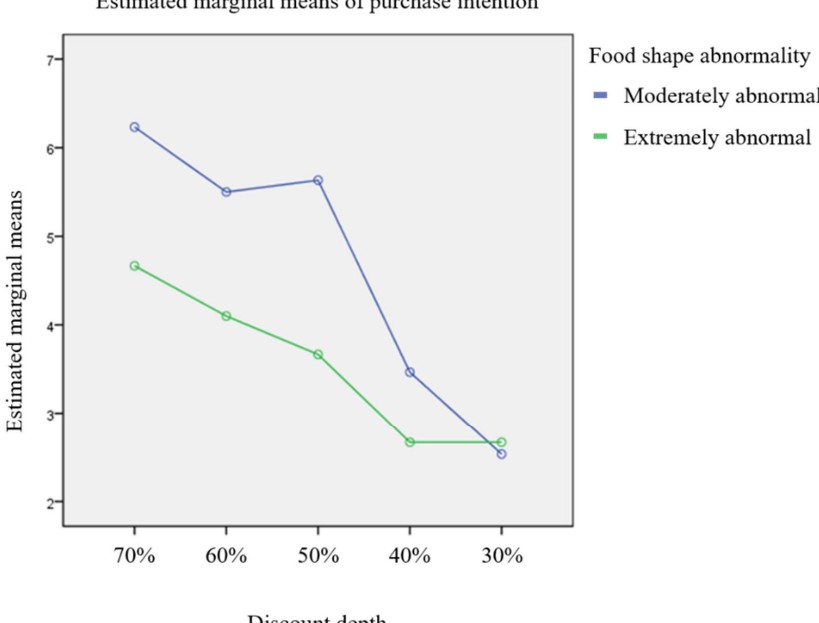

**Figure 3.** Effects of produce shape abnormality and discount on purchase intention.

## 5. Discussion

The results of Study 1 support Hypothesis 1, demonstrating that produce shape abnormality affects purchase intention. The consumers were least likely to purchase extremely abnormally shaped produce. The results indicate a significant difference in purchase intention between normal and moderately abnormal produce as well as between normal and extremely abnormal produce. A significant difference in purchase intention was also observed between moderately abnormal and extremely abnormal produce. These findings are consistent with those of Su et al. [18]. However, the results are inconsistent with those of Loebnitz et al. [7], which indicated no difference in purchase intention between normal and moderately normal products for Danish consumers. These results can be explained by assimilation–contrast theory, which suggests that assimilation occurs if the discrepancy between expected product performance and actual product performance is small. The Taiwanese consumers in this study likely perceived moderately abnormal produce to be different and of lower quality than normal produce, resulting in a significant difference in purchase intention.

Study 1 also revealed that the interaction between produce shape abnormality and organic labeling had a significant effect. Purchase intention was significantly higher for extremely abnormal produce with organic labeling. However, no significant difference in purchase intention was observed between normal or moderately abnormal produce with organic labeling and that without. The study supports the assumptions that intrinsic cues (i.e., food shape abnormality) are stronger than extrinsic cues (i.e., organic labeling) and that negative perceptions due to high-scope cues (produce shape) affect low-scope cues (organic labeling) by weakening their diagnostic power [7]. Studies have suggested that consumers' primary motivations for purchasing organic food products are related to health and taste [28,34,51]. One study noted that healthy ingredients strongly influenced purchase decisions [38]. Because organic products are generally healthier than nonorganic products, consumers find organic food more attractive. The results indicate that customers had the lowest intention to purchase extremely abnormally shaped produce without organic labeling; this result likely resulted from the perception that such products were of lower quality and less healthy.

The results from Study 2 support the assumption that consumers' intention to purchase moderately abnormal and extremely abnormal produce increases with discounts. These

results are consistent with those of other studies [48,49]. For moderately abnormal produce, discounts of 50% and higher should be offered to increase purchase intention among Taiwanese consumers. For extremely abnormal produce, a discount of 70% significantly affected purchase intention. Intention to purchase moderately abnormal produce was significantly higher than intention to purchase extremely abnormal produce at discounts of 40% or higher. This study contributes to the literature by determining the influence of price on the willingness of Taiwanese consumers to purchase imperfect food.

## 6. Conclusions

The results have several managerial implications. Intention to purchase abnormal produce increases with discounts, indicating that some Taiwanese consumers are willing to buy abnormally shaped produce at a reasonable price. This result is consistent with that of Loebnitz et al. [7], which suggested that having consumers experience unfamiliar food can help them accept abnormally shaped food. Our previous study also indicated a positive relationship between a pro-environmental identity and intention to purchase abnormally shaped foods [18]. Retailers should reconsider rejecting abnormally shaped food because of the growing number of individuals with a pro-environmental identity who may be willing to purchase them.

Certain companies in Taiwan donate unsold or unattractive perishable goods to nongovernmental groups, thereby fulfilling their social and environmental responsibilities. The New Taipei City Surplus Food Network provides an excellent model for other cities. The program collects unattractive vegetables, fruits, and other foods from local markets and sends them to the Social Welfare Department for distribution. Other social welfare groups and private organizations can implement similar programs to provide resources to those in need. Individual donations to facilitate the implementation of such programs should be encouraged. Food-saving and food-waste-reduction practices should be also be promoted through educational programs and various channels in the food industry (e.g., restaurants and businesses) to reduce food waste in Taiwan.

This study has several limitations. One is the use of convenience sampling during the data-collection process. Studies should use a larger sample to avoid nonprobability sampling and ensure the generalizability of the findings. Second, food shape abnormality was the only intrinsic cue; studies should examine whether other intrinsic cues of product quality, such as color, size, and texture, influence consumers' purchase intention. Studies should investigate other extrinsic cues in addition to organic labeling. Studies should also analyze the differences between consumers who are willing and unwilling to purchase abnormally shaped fruits and vegetables and the factors that influence the purchases of each consumer group. Studies on the effects of cultural and socioeconomic moderators may also yield valuable insights.

**Author Contributions:** Conceptualization, Y.-L.S.; methodology, Y.-L.S.; software, J.-H.W.; validation, P.N.; formal analysis, J.-H.W.; investigation, P.N. and J.-H.W.; resources P.N. and J.-H.W.; data curation, J.-H.W.; writing—original draft preparation, Y.-L.S.; writing—review and editing, Y.-L.S. and P.N.; visualization, P.N.; supervision, Y.-L.S.; project administration Y.-L.S. and P.N.; funding acquisition Y.-L.S. and P.N. All authors have read and agreed to the published version of the manuscript.

**Funding:** This research received no external funding.

**Institutional Review Board Statement:** Not applicable.

**Informed Consent Statement:** Not applicable.

**Data Availability Statement:** Not applicable.

**Acknowledgments:** This manuscript was edited by native English editors. All authors have consented to the acknowledgment.

**Conflicts of Interest:** The authors declare no conflict of interest.

## Appendix A

**Questionnaire A1.** Questionnaire example—Study 1.

The following picture is an organic certified moderately abnormal produce (50%)

1. Please indicate the abnormality of the food appeared above by circling the number on a 3-point scale (1 = "normal", 2 = "moderately abnormal", 3 = "extremely abnormal")

☐1        ☐2        ☐3

2. Is the food organic? ☐Yes   ☐No

3. How likely would you be to purchase these food items?" (1 = "very unlikely" and 7 = "very likely")

☐1        ☐2        ☐3        ☐4        ☐5        ☐6        ☐7

4. Gender: ☐Female        ☐Male

5. Age:

6. Education: ☐High school or less ☐Associate degree ☐Bachelor degree ☐Master degree or higher

**Questionnaire A2.** Questionnaire example—Study 2.

The following picture is a moderately abnormal produce (50%)

1. Please indicate the abnormality of the food appeared above by circling the number

(1 = "moderately abnormal", 2 = "extremely abnormal")

☐1        ☐2

2. How likely would you be to purchase these food items if the price is 40% off?

(1 = "very unlikely" and 7 = "very likely")

☐1        ☐2        ☐3        ☐4        ☐5        ☐6        ☐7

3. Gender: ☐Female        ☐Male

4. Age:

5. Education: ☐High school or less ☐Associate degree ☐Bachelor degree ☐Master degree or higher

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
