# Peer review of "Factors Affecting Taiwanese Consumers’ Intention to Purchase Abnormally Shaped Produce"

_sustainability, doi:10.3390/su15097715_

Round 1

Reviewer 1 Report

When reviewing scientific papers for publication, I usually start with a general overview in terms of a structure, abstract, literature review, methodology, findings of the research, discussion, conclusions, as well as limitations of the study.
In the assessment of the paper submitted for the review, I specifically focused on the discussed issues, applied research methods and the scope of analysis of research results, as well as substantive content of the article and its structure.
I do however have some remarks and feedback on the study.

1.First of all, it is suggested that authors supplement relevant questionnaire questions.

2.Explain the specific process of questionnaire collection, including time and so on.

3.How is the sample size determined? Whether it is supported by Gpower.

4.The specific process of research and analysis appears abstract because the questionnaire does not show it.

5.The authors themselves describe the convenience sampling results as less reliable, somewhat self-denying. Casts doubt on the reliability of the article.

Reviewer 2 Report

The manuscript addresses a relevant topic, especially in the context of the Thai market.

The introduction should explain the relevance of investigating the effect of organic labelling on the intention to purchase abnormally shaped produce.

The introduction should better explain the novelty of the study.

The literature review is appropriate to the research question, the hypotheses are logically derived and well supported in the literature.

The methodology and results sections of studies 1 and 2 are consistent in explaining how the research was conducted and the results obtained.

However, in section 3.1.2 it would be important to show pictures of products with the organic label.

I would also recommend including a table of means and standard deviations of purchase intention in Study 2 (similar to Table 1).

The conclusions should highlight the theoretical implications of the manuscript, as only management implications are described.

Reviewer 3 Report

This is a very interesting research on the influence of shape abnormality, organic labeling, and dis-12 counts on consumers’ intention to purchase. Authors perform a clear research based on a convenience sample, while stating the limitations of this.

I suggest to explain more clearly the results of the two-way analysis of variance in Table 2, as it is not intuitive the intrepretation of "corrected model" and "intercept" rows. Maybe you should include a citation to the two-way anova method.

Round 2

Reviewer 1 Report

I think the authors have answered my questions.